# Home Cooking and Child Obesity in Japan: Results from the A-CHILD Study

**DOI:** 10.3390/nu11122859

**Published:** 2019-11-21

**Authors:** Yukako Tani, Takeo Fujiwara, Satomi Doi, Aya Isumi

**Affiliations:** Department of Global Health Promotion, Tokyo Medical and Dental University (TMDU), 1-5-45, Yushima, Bunkyo-ku, Tokyo 113-8519, Japan; tani.hlth@tmd.ac.jp (Y.T.); doi.hlth@tmd.ac.jp (S.D.); isumi.hlth@tmd.ac.jp (A.I.)

**Keywords:** home cooking, meal preparation, obesity, children, parenting

## Abstract

This study aimed to investigate the association between the frequency of home cooking and obesity among children in Japan. We used cross-sectional data from the Adachi Child Health Impact of Living Difficulty study, a population-based sample targeting all fourth-grade students aged 9 to 10 in Adachi City, Tokyo, Japan. Frequency of home cooking was assessed by a questionnaire for 4258 caregivers and classified as high (almost every day), medium (4–5 days/week), or low (≤3 days/week). School health checkup data on height and weight were used to calculate body mass index *z*-scores. Overall, 2.4% and 10.8% of children were exposed to low and medium frequencies of home cooking, respectively. After adjusting for confounding factors, children with a low frequency of home cooking were 2.27 times (95% confidence interval: 1.16–4.45) more likely to be obese, compared with those with a high frequency of home cooking. After adjustment for children’s obesity-related eating behaviors (frequency of vegetable and breakfast intake and snacking habits) as potential mediating factors, the relative risk ratio of obesity became statistically non-significant (1.90; 95% confidence interval: 0.95–3.82). A low frequency of home cooking is associated with obesity among children in Japan, and this link may be explained by unhealthy eating behaviors.

## 1. Introduction

The prevalence of childhood overweight and obesity has increased worldwide in recent decades, becoming a major public health epidemic [1]. In Organization for Economic Cooperation and Development (OECD) countries, the prevalence of childhood overweight and obesity is 25% [2]. Obese children are more likely to become obese adults [3] and are at a higher risk of a wide range of serious health complications [4]. To combat the obesity epidemic, it is important that prevention efforts are shaped by a solid evidence base regarding the risk factors for obesity in children.

Home cooking has been suggested as a key strategy to prevent obesity [5]. In developed countries, sociodemographic changes, such as an increasing numbers of working women and single-parent or small families, have led to less time being available for home cooking and an increased shift toward eating out or buying prepared meals [6,7,8]. In Japan, household expenditure for prepared food has increased in recent decades, and eating out is common among younger age groups [9]. Increased consumption of out-of-home foods, such as fast food and convenience food, is a major concern because these foods are higher in calories, fat, and sodium; lower in fiber and calcium [10,11,12,13]; and associated with poor diet quality and increased energy intake among children [12,13,14,15]. Several studies conducted in the United States have shown an association between fast food consumption and increased weight gain in adolescents [16,17]. However, these studies evaluated only the effect of eating out; they did not examine the effect of at-home consumption of prepared (precooked) meals, such as packed lunches, convenience foods, and ready-made meals. To account for the effects of both eating out and eating prepared meals at home, it is necessary to examine the frequency of home cooking.

Several studies of adults have suggested that home cooking improves diet quality and weight status [18,19,20]. However, there is limited evidence on the relationship between home cooking and children’s weight status. Parents who report a low enjoyment of cooking, little meal planning, and fewer hours spent on food preparation are less likely to serve vegetable and fruits, whereas they are more likely to serve fast food for family dinners [21]. Furthermore, because children may model their parents’ healthy eating habits, enjoy meal time more, and feel closer to their caregivers through having home-cooked meals, home cooking may be beneficial for healthy child development, including weight status, because it fosters a better relationship between the caregiver and the child [22,23,24,25,26]. Moreover, because dietary intake patterns established early in life tend to persist into adolescence and adulthood [27,28,29], it is also important to understand the associations between home-cooking habits and child weight status. Children who eat less home-cooked food may be vulnerable to the development of unhealthy dietary behaviors, which may, in turn, be linked to obesity. However, to the best of our knowledge, no studies have been conducted to examine the association between home cooking and child obesity in Japan.

The purpose of the present study was to examine the association between the frequency of home cooking and obesity among children in Japan.

## 2. Materials and Methods

### 2.1. Study Design and Subjects

The Adachi Child Health Impact of Living Difficulty (A-CHILD) project was established in 2015 to evaluate the determinants of health among children in Adachi City, Tokyo, Japan [30,31,32]. We used data from the 2018 wave of the A-CHILD study. The survey covered all 69 public elementary schools established in Adachi City, Tokyo, Japan. In 2018, self-reported questionnaires with anonymous unique identification numbers were distributed to 5311 children in the fourth grade (aged 9–10 years) in these elementary schools. The children were asked to pass on the questionnaires to their caregivers at home to complete. The children then returned the completed questionnaires to their schools. A total of 4605 caregivers completed the questionnaires (response rate: 86.7%), and 4290 provided informed consent and returned all the questionnaires. To examine child body mass index (BMI), we used school health checkup data on body height and weight, collected in April or May 2018, which was linked to the questionnaire responses using the anonymous unique identification number. Participants who did not indicate their body height, weight status, or month of birth (*n* = 20) were excluded from the analysis, as were participants who did not complete the questions related to home-cooking status (*n* = 12). After these exclusions, 4258 participants were included in the study. The final sample comprised information on 2151 boys and 2107 girls. Of the caregivers who completed the questionnaires, 91.5% were mothers and 7.7% were fathers. The A-CHILD protocol and the use of the data in this study were approved by the Ethics Committee at Tokyo Medical and Dental University (No. M2016-284).

### 2.2. Child Body Height and Weight Status

School teachers assessed child height and weight at elementary schools during a school health checkup, following a standardized protocol [33]. Height was measured to the nearest 0.1 cm using a portable stadiometer, and weight was measured to the nearest 0.1 kg on a digital scale, without shoes and in light clothing. BMI was calculated by dividing the child’s weight (in kilograms) by the square of body height (in meters). BMI was expressed as a *z*-score representing the deviation in standard deviation units from the mean of a standard normal distribution of BMI specific to age and sex, according to the World Health Organization’s Child Growth Standards. Children’s BMIs were categorized as underweight/normal weight (<+1 SD), overweight (≥+1 SD and <+2 SD), or obese (≥+2 SD) using standard deviation cut-off points [34].

### 2.3. Home-Cooking Frequency

Following previous studies [35,36], home-cooking frequency over the past month was assessed using the following question: “How many times did you or someone else in your family cook meals at home? Include a simple meal, such as fried eggs, as a cooked meal”. The five response categories were almost every day, 4–5 days/week, 2–3 days/week, a few days/month, and rarely. Considering the distribution of the answers to this question and based on categories previously used in other studies [35,36], we collapsed the responses into three groups of frequency of home cooking: (i) High (almost every day: 86.8%); (ii) medium (4–5 days/week: 10.8%); and (iii) low (≤3 days/week: 2.4%).

### 2.4. Covariates

Possible covariates, such as children’s physical activity and eating behaviors, were also assessed using the caregiver-completed questionnaires. Children’s physical activity was assessed using the frequency of physical activity for 30 min or more during the week (never/rarely, 1–2 times/week, 3–4 times/week, or ≥5 times/week). Children’s eating behavior included frequency of vegetable intake (twice/day, once/day, or <3 times/week), frequency of breakfast intake (every day, often, or rarely/never), and snacking habits (no snacking, snacking at a set time (controlled), or snacking freely) [32]. Household characteristics included the caregiver’s marital status (married or living with partner, single, divorced, or widowed), the child having siblings (yes or no), cohabitation with the child’s grandparents (yes or no), and annual household income (<3.00, 3.00–5.99, 6.00–9.99, or ≥10.0 million yen). Caregiver characteristics included mother’s age (<35, 35–44, or ≥45 years), mother’s educational attainment (low (junior high school, dropped out of high school, or completed high school), middle (professional school, some college, or dropped out of college), or high (completed college or more), mother’s employment and time of returning home from work (employed/returning home before 18:00, employed/returning home 18:00–20:00, employed/returning home after 20:00, employed/returning home at irregular times, or not employed), and mother’s and father’s BMIs calculated using self-reported height in centimeters and weight in kilograms. Standard categories of BMI [37] were used to characterize parents as obese (BMI ≥ 30.0 kg/m^2^), overweight (BMI = 25.0–29.9 kg/m^2^), normal weight (BMI = 18.5–24.9 kg/m^2^), or underweight (BMI < 18.5 kg/m^2^).

### 2.5. Statistical Analysis

First, the participants’ characteristics were stratified by home-cooking status, and differences were tested using Pearson’s chi-square test. Second, we calculated relative risk ratios (RRRs) and 95% confidence intervals (CIs) of overweight and obesity using multinomial logistic regression. Two models were constructed for both overweight and obesity. Model 1 adjusted for child’s sex, physical activity, household characteristics (marital status, having siblings, living with grandparents, and household income), and caregiver characteristics (mother’s age, education, employment, and BMI and father’s BMI) as potential confounders. Model 2 additionally adjusted for the child’s obesity-related eating behaviors (frequency of vegetable intake, frequency of breakfast consumption, and snacking habits) as potential mediating factors because children may learn to model their eating behaviors based on their caregivers through home cooking, which may affect children’s weight status.

## 3. Results

The majority of the caregivers cooked at home for their children almost every day (86.8%), whereas 2.4% of the caregivers cooked at home less often than 3 days per week and 10.8% cooked at home 4 to 5 days per week. The breakdown of the distribution of cooking at home less often than 3 days per week was 1.8% for 2 to 3 days/week, 0.5% for a few days/month, and 0.1% for rarely. Overall, 14.7% of the children were overweight, and 5.9% were obese (Table 1). In terms of household status, 72.8% of the caregivers were married, 80.5% of the children had siblings, 10.2% lived with the child’s grandparents, and 10.7% were poor households with annual incomes of less than three million yen. (Table 2) The most common maternal education level was professional school, some college, or dropped out of college. In 21% of the households, the mothers returned home from work after 18:00 or irregularly. When the mothers did not work or returned home from work before 18:00, the frequency of home cooking was high. The frequency of home cooking was low in households with a non-married parent and in those with low income. Children exposed to a low frequency of home cooking tended to have lower frequencies of vegetable and breakfast intake and snacked freely (Table 1).

After adjusting for potential confounding factors, children who were exposed to a low frequency (≤3 days/week) of home cooking were 2.27 times (95% CI: 1.16–4.45) more likely to be obese, compared with children who were exposed to home cooking almost every day (Table 3, Model 1). After adjustment for children’s obesity-related eating behaviors as potential mediators, this RRR was reduced and became statistically non-significant (*RRR* = 1.90; 95% CI: 0.95–3.82) (Table 3, Model 2).

## 4. Discussion

To our knowledge, this is the first study to examine the association between the frequency of home cooking and obesity among children aged 9 to 10 years. We found that a low frequency (<3 days per week) of home cooking doubled the risk of obesity for children, even after controlling for child’s sex, physical activity, household characteristics (parents’ marital status, siblings, living with grandparents, and household income), and parents’ individual characteristics (maternal age, education, and employment, and BMI for both parents). This association was attenuated after controlling for potential mediating factors (i.e., child’s obesity-related eating behaviors), suggesting that children’s eating behaviors partially mediated the association between home cooking and children’s obesity.

Three possible factors may explain the link between less frequent home cooking and child obesity: (i) Caregivers’ food choices; (ii) healthy eating practices; and (iii) children eating similar foods to those eaten by their caregivers. These potential mechanisms could play a role in determining whether the association between home cooking and obesity in children is direct (i.e., home cooking is directly associated with child obesity) or indirect (i.e., home cooking influences children’s eating behavior, which, in turn, is associated with child obesity). We found that the association between home cooking and the child’s obesity became non-significant after adjusting for the child’s frequency of vegetable and breakfast intake and snacking habits in Model 2. This finding may be explained by the caregiver’s food choice: Caregivers who usually cook at home may be more likely to select healthier foods, compared with caregivers who provide out-of-home food. Consistent with this idea, a previous study found that home cooking was associated with higher vegetable consumption among children [38]. Two other studies demonstrated that, among adults, a higher frequency of home-cooked meals was associated with indicators of a healthier diet, including fruit and vegetable intake, Mediterranean diet score, and Dietary Approaches to Stop Hypertension (DASH) score [20,39]. Compared with food prepared at home, food prepared outside of the home is higher in calories as well as total and saturated fats and has less fiber, calcium, and iron [10]. Alexy et al. argued that convenience foods have a high fat content and contain many flavorings and food additives [12]. Previous research has suggested that the presence of children in the household might be protective for family body weight: Sobal et al. reported an inverse association between the frequency of family meals and body weight for adults with children at home, but no such association was found among adults without children [40]. These results may indicate that cooking at home with children is beneficial for creating a healthy food environment.

A second explanation is that the frequency of home cooking may be a proxy for caregivers’ healthy eating practices. For example, caregivers who cook at home infrequently might be less likely to prepare breakfast for their children, which is associated with an increased risk of children becoming obese [41,42]. This is supported by our finding that children who have a low frequency of home-cooked meals are more likely to skip breakfast and to eat snacks freely (Table 1). We also found that the significant association between home cooking and obesity disappeared after adjusting for children’s eating behaviors, including skipping breakfast and snacking habits (Table 3, Model 2). In previous systematic reviews examining the association between parental practices and children’s consumption of unhealthy foods (including snacks and sugar-sweetened beverages), restrictive parental guidance/rulemaking and control of the availability of unhealthy foods were the practices that were most positively associated with children’s consumption of unhealthy foods [43,44]. Children who snack frequently have been shown to consume higher total energy and energy from sugars [45]. Therefore, a low frequency of home cooking may be a proxy for less effective parental practice in terms of children eating healthy foods, which could explain the association with obesity in children.

A third potential explanation for this association is that home cooking may lead to a healthy diet because children eat foods similar to those consumed by their parents. When children eat home-cooked meals, they tend to eat the same foods as their parents. In contrast, when children eat outside the home or consume prepared meals, they select what they want and thus eat different foods from those eaten by their parents. Previous studies have reported that children who eat similar foods to those eaten by their parents are more likely to have healthy diets [38,46], suggesting that children may miss out on specific nutrients or food types, such as vegetables, if they are served a separate “child meal”. Home cooking may also create a supportive and positive food environment for children. Creating a positive atmosphere at mealtime supports children’s opportunities to try new foods and to develop their own food preferences [22].

Several limitations of this study should be mentioned. First, we assessed the frequency of cooking using a simple questionnaire based on previous studies, but the validity and reliability of the questionnaire were not examined here or in these existing studies. However, we confirmed the plausibility of the results by testing the association of the frequency of home cooking with the child’s vegetable intake and breakfast skipping. Second, we defined home cooking as a basic and simple practice, such as frying an egg, and we did not assess the quality of the meals being prepared. Therefore, caregivers who cook low-quality meals (e.g., meals with little variety or unhealthy meals) may be included in the high frequency of home cooking category. This may have led to an underestimate of the association between home cooking and children’s obesity. However, in large-scale surveys, it is difficult to evaluate the quality of meals because a great deal of time would be required for this assessment. Additionally, our study focused on parents engaging in the behavior of preparing meals for their children rather than on the quality of the meals they prepared. Third, we did not account for caregivers’ food knowledge, which is particularly important for preventing obesity in children; this topic warrants further research. Fourth, because our sample of school children was from only one city, the generalizability of the results may be low. Furthermore, the caregivers in the present study were well educated, and most of the respondents were mothers. Low parental socioeconomic status was linked to children’s obesity, and we also found that low maternal educational attainment was significantly associated with children’s obesity (data not shown); this may have led to an underestimate of the association between home cooking and children’s obesity. Finally, because this was a cross-sectional study, we were unable to assess causality. Longitudinal studies or randomized controlled trials are needed in the future to clarify the effectiveness of home cooking in preventing obesity in children. Despite these limitations, we were able to demonstrate a significant association between a low frequency of home cooking and children’s obesity, controlling for potential confounding factors, and our findings may be useful for identifying potential targets for interventions aimed at improving children’s body weight management.

## 5. Conclusions

Our study has provided novel findings regarding the association between home cooking and children’s body weight status. Home cooking presents an opportunity for parents to offer a model of healthy eating to their children and to pass on food traditions from their own culture. When children participate in meal preparation, they tend to eat healthy diets [47]. Future studies are needed to clarify the causal relationship and mechanisms through which home cooking influences children. Although the present study focused on body weight, future studies should also examine other physical, psychological, and social outcomes that may be associated with home cooking for children.

## Figures and Tables

**Table 1 nutrients-11-02859-t001:** Characteristics of participating children (*n* = 4258).

	Total		Frequency of Home Cooking			χ^2^	*p*-Value
			High (*n* = 3695, 86.8%)	Medium (*n* = 461, 10.8%)	Low (*n* = 102, 2.4%)		
	*n*	%	%	%	%		
Child’s status							
Sex							
Boy	2164	50.6	51.1	47.1	46.6	3.4	0.18
Girl	2114	49.4	48.9	52.9	53.4		
Body weight status (BMI for age *z*-score)							
Normal weight/ underweight (<1 SD)	3384	79.5	80.1	76.8	67.6	11.8	0.003
Overweight (≥1 SD and <2 SD)	624	14.7	14.3	15.8	20.6		
Obese (≥2 SD)	250	5.9	5.5	7.4	11.8		
Physical activity							
Never/rarely	501	11.7	10.8	16.3	25.2	38.8	<0.001
1–2 times/week	1600	37.4	37.5	38.1	32.0		
3–4 times/week	1189	27.8	28.6	22.2	23.3		
≥5 times/week	969	22.7	22.7	23.0	17.5		
Missing	19	0.4	0.4	0.4	1.9		
Eating behaviors							
Frequency of vegetable intake							
Twice/day	1748	40.9	43.6	22.8	24.3	259.6	<0.001
Once/day	2072	48.4	48.2	54.0	31.1		
<3 times/week	446	10.4	8.0	22.8	42.7		
Missing	12	0.3	0.2	0.4	1.9		
Frequency of breakfast intake							
Every day	3822	89.3	91.1	78.7	75.7	113.2	<0.001
Often	330	7.7	6.4	17.2	13.6		
Rarely/never	82	1.9	1.6	2.8	9.7		
Missing	44	1.0	1.0	1.3	1.0		
Snacking habits							
Not snacking	278	6.5	6.4	6.5	8.7	43.2	<0.001
Snacking at a set time (controlled)	1856	43.4	45.3	31.4	29.1		
Snacking freely	2104	49.2	47.4	61.1	61.2		
Missing	40	0.9	0.9	1.1	1.0		

BMI: body mass index; SD: standard deviation.

**Table 2 nutrients-11-02859-t002:** Characteristics of participating households and caregivers (*n* = 4258).

	Total		Frequency of Home Cooking			χ^2^	*p*-Value
			High	Medium	Low		
			(*n* = 3695, 86.8%)	(*n* = 461, 10.8%)	(*n* = 102, 2.4%)		
	*n*	%	%	%	%		
Household status							
Marital status							
Married/common-law marriage	3098	72.8	74.6	61.8	53.9	84.7	<0.001
Single/divorced/widowed	352	8.3	7	15	24.5		
Other/missing	808	19	18.4	23.2	21.6		
Other children in the household							
No	836	19.5	18.1	27.7	35.0	39.9	<0.001
Yes	3442	80.5	81.9	72.3	65.0		
Living with the child’s grandparents							
No	3840	89.8	89.4	92.7	90.3	4.7	0.10
Yes	438	10.2	10.6	7.3	9.7		
Household income (million yen)							
<3.00	457	10.7	9.7	15.7	22.3	35.7	<0.001
3.00–5.99	1275	29.8	29.7	30.1	33.0		
6.00–9.99	1411	33.0	34.0	27.3	20.4		
≥10.0	499	11.7	11.8	11.0	9.7		
Missing	636	14.9	14.7	15.9	14.6		
Caregiver’s status							
Mother’s age (years)							
<35	470	11.0	10.2	15.3	18.4	26.6	<0.001
35–44	2573	60.1	60.7	57.6	51.5		
≥45	1098	25.7	26.1	23.2	22.3		
Missing	137	3.2	3.0	3.9	7.8		
Mother’s education							
Low	1098	25.7	25.0	29.0	34.0	33.2	<0.001
Middle	1381	32.3	33.2	26.9	24.3		
High	646	15.1	15.8	10.5	8.7		
Other/missing	1153	27.0	26.0	33.5	33.0		
Mother’s employment and time of returning home from work							
Employed/returning home before 18:00	1949	45.8	47.1	37.3	35.3	74.0	<0.001
Employed/returning home 18:00–20:00	622	14.6	13.8	20.2	17.6		
Employed/returning home after 20:00	143	3.4	2.8	6.3	11.8		
Employed/returning home irregularly	136	3.2	3.1	4.3	2.9		
Not employed	928	21.8	22.4	18.0	15.7		
Missing	480	11.3	10.8	13.9	16.7		
Parents’ BMI							
Mother’s BMI							
Underweight (<18.5 kg/m^2^)	487	11.4	11.5	11.2	7.8	13.4	0.10
Normal weight (18.5–24.9 kg/m^2^)	2817	65.8	66.4	61.7	63.1		
Overweight (25.0–29.9 kg/m^2^)	428	10.0	9.7	12.5	8.7		
Obese (≥30 kg/m^2^)	89	2.1	2.0	2.2	2.9		
Missing	457	10.7	10.3	12.5	17.5		
Father’s BMI							
Underweight (<18.5 kg/m^2^)	63	1.5	1.3	2.4	3.9	35.3	<0.001
Normal weight (18.5–24.9 kg/m^2^)	2286	53.4	54.7	45.8	41.7		
Overweight (25.0–29.9 kg/m^2^)	909	21.2	21.2	21.9	21.4		
Obese (≥30 kg/m^2^)	169	4.0	4.1	3.0	2.9		
Missing	851	19.9	18.7	26.9	30.1		

BMI: body mass index.

**Table 3 nutrients-11-02859-t003:** Adjusted relative risk ratios of overweight and obesity according to the frequency of home cooking among school children in Japan (*n* = 4258).

	Overweight		Obesity	
	RRR (95% CI)	*p*-Value	RRR (95% CI)	*p*-Value
Model 1				
Frequency of home cooking				
High (Almost every day)	ref		ref	
Medium (4–5 times/week)	1.07 (0.81–1.41)	0.60	1.33 (0.89–1.99)	0.20
Low (≤3 times/week)	1.46 (0.87–2.46)	0.16	**2.27 (1.16–4.45)**	**0.02**
Model 2				
Frequency of home cooking				
High (Almost every day)	ref		ref	
Medium (4–5 times/week)	1.09 (0.82–1.44)	0.52	1.31 (0.87–1.98)	0.24
Low (≤3 times/week)	1.46 (0.86–2.48)	0.17	1.90 (0.95–3.82)	0.08

RRR: relative risk ratio; CI: confidence interval; ref: reference. Boldface indicates statistical significance (*p* < 0.05). Model 1 adjusted for child’s sex, physical activity, household status (parents’ marital status, siblings, living with grandparents, and household income), and caregiver’s status (mother’s age, education, employment, and BMI and father’s BMI). Model 2 adjusted for child’s eating behaviors (frequency of vegetable and breakfast intake and snacking habits), as well as all variables in Model 1.

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
