# Peer review of "Home Cooking and Child Obesity in Japan: Results from the A-CHILD Study"

_nutrients, 2019, doi:10.3390/nu11122859_

Round 1

Reviewer 1 Report

In this post-hoc analysis of a large cohort of Japanese children, the results of a questionnaire including some simple questions regarding home cooking and eating habits, a very weak association is shown between limited home-cooking (<3 times/week) and obesity. This association no longer is significant when controlled for eating habits (such as snacking).

Although the authors mention in their discussion that one can not conclude a "causative relation" when simply descriging an assocition, the abstract concludes with the statement that promotion of home-cooking may be beneficial in the prevention of obesity. The method is incompletely described. There is no information on validation of the questionnaire. It is not clear whether the study was designed from the start to look for the effects of home-cooking (and if so, why the questions did not include items regarding the quality of the prepared food?), or is it a post-hoc analysis. Statistics are not clear to me, why were 4 models not combined in the multiple regression model? While "model 4" shows that the relation of obesity with home-cooking is not longer significant when controlled for eating behavior, the discussion is also solely centered around home-cooking, and does not give attention to model 4. The discussion insufficiently highlights the limitations of this study.

Overall Englisch language needs correction.

Reviewer 2 Report

Dear Authors,

The topic of this manuscript is very topical and could be of interest to the readership. Please see below my comments that I feel will help improve the readability and impact of the manuscript.

Abstract:

Please add results of fourth model that adjusts for child eating behaviours as it is currently misleading.

Manuscript in general:

Please edit for language, grammatical errors and sentence structuring, some examples provided in following comments.

Introduction:

Line 31: change 'risk factors of obese in children' to 'risk factors of obesity in children'

Line 42 - 44: Consider restructuring sentence

Line 43: change to 'meals'

Line 44: what is instant food, do you mean ready-made meals, if not please provide examples, and is delicatessen - delicatessen meats (e.g. processed meats), please further explain.

Line 45: please change to 'prepared meals,' '....to examine home cooking.'

Lines 46 - 48: Sentence restructuring, perhaps consider: 'Several studies in adults have suggested that home cooking improves diet quality and potentially contributes to a lower BMI, however, few studies have been conducted on the impact of home cooking on children's weight status.

Lines 48 - 49: Sentence needs rewriting

Lines 55 - 56: Please be careful of using definitive language without referencing, consider changing to 'Children who eat less home-cooked food may be vulnerable to developing unhealthy dietary behaviours, which in turn may be linked to obesity.'

Methods:

Line 65: Change 'performed' to 'conducted'

Line 69: Change 'fill out' to 'complete'

Lines 74 - 75: 'because... z scores' not needed

Lines 93 - 94: Were any other example of what was considered to be a 'home cooked' meal given?

Lines 95 - 96: 'The distribution...' belongs in results

Results:

Lines 137 - 138: Is this medium level? It seems quite high

Table 1: Please split into a Table for children and one for the caregivers as it is currently very big which may make it difficult to follow for the readers

Table 1: In mother's employment, rephrase 'go home time'

Table 1: Parental BMI - change how it is written to Underweight (<18.5Kg/m2); Normal weight (18.5 - 24.9 Kg/m2); Overweight (25.0 - 29.9 Kg/m2); Obese (≥30.0 Kg/m2), in all instances

Lines 146 - 156: Please present p-values when discussing significance levels or include in Table 2

Discussion:

Please proof read all for English, language and sentence structuring

Lines 196 - 198: Just because they do not cook at home frequently does not mean that they do not use portion control, please remove sentence or add reference

Lines 199 - 201: 'poor parental interest on food for children... and thus associated with obesity of children' needs rephrasing and some referencing, this is shaming language and could be considered offensive to many people, just because they do not cook at home does not mean they are not interested in the food they provide for their children, they may have low skill levels, poor cooking facilities, time restrictions. There are a number of examples were the language is shaming parents which some of the readers may find offensive and I would consider rephrasing

Limitations:

Additional limitations should be included such as the generalisability of the study, that the majority of the sample were female, that the sample was fairly well educated.

Lines 219 - 221: a low frequency of home cooking was not associated with child's BMI when the model was adjusted for the child's eating behaviours, please change this

Lines 225 - 226: For the majority of people, their working times are not flexible and thus logistically this seems like an impractical recommendation, I would suggest removing it

Lines 226 - 227: This is a very interesting point, and could be expanded about the potential of including children in the entirety of the cooking process

Overall this is an interesting piece of research.

Reviewer 3 Report

Childcare environments are critical spaces to shape healthy eating practices that track throughout the lifespan. Understanding key determinants in a child’s environment that influence these outcomes is important for intervention planning and education. The home environment is the primary space for learning about food habits. Food prepared in the home has been demonstrated to be of higher nutritional value compared with food prepared away from home.

A cross-sectional study assessing relative risk ratios to investigate this phenomenon is appropriate to the research question of identifying associations between food cooked in the home and weight status of children. A descriptive study can provide some base associations that can be used to build intervention studies to explore solutions to the problem.

Some general concerns with this study include the writing, citations, and lack of alignment of study design with study conclusions. First, the writing and grammar within this paper requires attention and makes reading and interpreting this article difficult. Secondly, the citations provided within the text are older. I encourage you to update the information cited. There has been a systematic review in this area published by Simmonds and colleagues 2016 which would inform the paper. Lastly, conclusion statements in the paper use language of preventive and benefit. Considering the aim of the study is 'association', the conclusion and discussion statements should align with the study design. 

Generally, citations throughout the paper would benefit from including more current information in the literature.

Line 22 the conclusion statement in the abstract indicates that home cooking may be a ‘preventative measure in obesity management’ which is in direct contradiction with the study purpose inline 10 which states that the aim of the study was to investigate ‘associations’ between home cooking and obesity. The language of ‘preventative’ indicates a cause-and-effect connection.

Line 42-45 Sentence structure and verb tense make this difficult to read and thus understand. This occurs throughout the paper. Review the entire paper and revise it for grammar.

Line 47 Home cooking is associated with improved reporting of weight status but does not ‘make BMI lower’

Line 78 Respondents and sample unclear in this sentence. Clarify

Line 95-96 Results should be reported in the results section. The distribution of responses should be shifted to the results

Line 121 states data ‘were tested using Chi-square tests’; provide specifics on the type of Chi-square test used.

Line 139-141 Concept or intended message is unclear

Line 223-224 the conclusion statement that encouraging caregivers to cook at home may be beneficial in managing the child’s body weight. This is indicative of a cause-and-effect conclusion that is not supported by the results. Align the statement so that the study design is reflected in the conclusion statement.

Round 2

Reviewer 1 Report

The authors have adequately adressed the remarks in the first review, and adapted the discussion and conclusion accordingly. My only remark concerns the questionnaire, which also in the original publications, does not seem to have been validated (as is often the case in this kind of studies, but it should be mentioned)
